# Gut Microbiota Dysbiosis in COVID-19: Modulation and Approaches for Prevention and Therapy

**DOI:** 10.3390/ijms241512249

**Published:** 2023-07-31

**Authors:** Virna Margarita Martín Giménez, Javier Modrego, Dulcenombre Gómez-Garre, Walter Manucha, Natalia de las Heras

**Affiliations:** 1Instituto de Investigaciones en Ciencias Químicas, Facultad de Ciencias Químicas y Tecnológicas, Universidad Católica de Cuyo, San Juan 5400, Argentina; vmartin@uccuyo.edu.ar; 2Laboratorio de Riesgo Cardiovascular y Microbiota, Hospital Clínico San Carlos-Instituto de Investigación Sanitaria San Carlos (IdISSC), 28040 Madrid, Spain; javier.modrego@salud.madrid.org; 3Centro de Investigación Biomédica en Red Enfermedades Cardiovasculares (CIBERCV), Instituto de Salud Carlos III, 28029 Madrid, Spain; 4Departamento de Fisiología, Facultad de Medicina, Plaza Ramón y Cajal, s/n. Universidad Complutense, 28040 Madrid, Spain; 5Área de Farmacología, Departamento de Patología, Facultad de Ciencias Médicas, Universidad Nacional de Cuyo, Mendoza 5500, Argentina; wmanucha@fcm.uncu.edu.ar; 6Instituto de Medicina y Biología Experimental de Cuyo (IMBECU), Consejo Nacional de Investigaciones Científicas y Tecnológicas (CONICET), Mendoza 5500, Argentina

**Keywords:** gut microbiota, inflammation, immune system, COVID-19, SARS-CoV-2, probiotics

## Abstract

Inflammation and oxidative stress are critical underlying mechanisms associated with COVID-19 that contribute to the complications and clinical deterioration of patients. Additionally, COVID-19 has the potential to alter the composition of patients’ gut microbiota, characterized by a decreased abundance of bacteria with probiotic effects. Interestingly, certain strains of these bacteria produce metabolites that can target the S protein of other coronaviruses, thereby preventing their transmission and harmful effects. At the same time, the presence of gut dysbiosis can exacerbate inflammation and oxidative stress, creating a vicious cycle that perpetuates the disease. Furthermore, it is widely recognized that the gut microbiota can metabolize various foods and drugs, producing by-products that may have either beneficial or detrimental effects. In this regard, a decrease in short-chain fatty acid (SCFA), such as acetate, propionate, and butyrate, can influence the overall inflammatory and oxidative state, affecting the prevention, treatment, or worsening of COVID-19. This review aims to explore the current evidence regarding gut dysbiosis in patients with COVID-19, its association with inflammation and oxidative stress, the molecular mechanisms involved, and the potential of gut microbiota modulation in preventing and treating SARS-CoV-2 infection. Given that gut microbiota has demonstrated high adaptability, exploring ways and strategies to maintain good intestinal health, as well as an appropriate diversity and composition of the gut microbiome, becomes crucial in the battle against COVID-19.

## 1. Introduction

COVID-19 (by the acronym of Coronavirus Disease of 2019) is a respiratory disease caused by a novel coronavirus (SARS-CoV-2, by the abbreviation of Severe Acute Respiratory Syndrome Coronavirus 2) that continues to affect millions of people worldwide. While most COVID-19 patients experience respiratory symptoms, up to 20% exhibit gastrointestinal symptoms, such as diarrhea [1], suggesting that the gastrointestinal tract is an additional site of SARS-CoV-2 infection apart from the lungs. 

SARS-CoV-2 enters host cells by using the angiotensin-converting enzyme 2 (ACE2) receptor, which is highly expressed in both the respiratory and gastrointestinal tracts. Consequently, ACE2 plays a significant role in regulating intestinal inflammation and the microbial ecology of the gut.

Since the gastrointestinal tract serves as the largest immune organ in humans and plays a critical role in defending against infections by pathogens, it is crucial to comprehend the impact of SARS-CoV-2 infection on the host’s gut microbiota and its potential long-term effects on human health.

## 2. General Concept of Gut Microbiota in a Healthy State and in Dysbiosis

The microbiota is a community of live microorganisms (bacteria, viruses, fungi, archaea and protozoa) that inhabit humans and help us maintain homeostasis [2]. This mutually beneficial relationship is so close that we are considered as a “holobiont”, a superorganism made up of the host and the microorganisms that live in symbiosis with it [3]. The concept of microbiome refers to the set of microorganisms, their genes, and derived metabolites in an ecosystem. The human being has about 23,000 genes, while the microbiota contributes a number 150 times greater. Human microbiota consists of the 10–100 trillion symbiotic microbial cells harbored by each person, mainly bacteria [4,5]. Most of these are in the gut, and more than 90% of the bacteria of our entire organism are found in the colon only [6]. These data are important considering that most of the studies are carried out on fecal samples, which mainly represent the distal intestinal (colonic) microbiota. Thus, fecal specimens are naturally collected, may be sampled repeatedly, and constitute a less invasive procedure than others, such as intestinal biopsy sample collection, luminal brushing, and intestinal fluid aspirate, which are not suitable methods for healthy people [7,8]. However, it is important to clarify that stool samples do not represent the totality of the microbiota adhered to the intestinal epithelium and that the bacteria of the upper intestinal tracts are not correctly detected [9].

The conformation of the microbiota in each subject is unique and is determined by a multitude of factors, such as the way we are born, whether it is by vaginal delivery or caesarean section [10], the type of diet [11], drug intake (mainly antibiotics) [12], and even if one lives in an urban or rural environment [13].

The bacteria that predominate in the intestinal microbiota belong to the Bacteroidetes and Firmicutes phyla [14], and in most individuals, the microbiota could be classified into one of three enterotypes according to the dominant genera *Bacteroides*, *Prevotella* or *Ruminococcus* [15], mainly by the effects of diet [16]. Gut microbiota has a profound influence on human physiology and nutrition, and it has been demonstrated to be essential for human life [17]. The more different types of bacteria in the gut and the more evenly distributed they are, the greater the diversity. A diverse microbiome can perform many more functions, making the whole system more stable. In fact, a diverse and balanced intestinal microbiota ensures the correct functioning of the digestive tract, strengthens the immune system, and improves metabolism [18,19]. 

Gut microbiota performs a wide variety of biochemical and physiological functions that influence the host’s metabolism [20]. The microorganisms that compose it have various enzymes that make it possible to transform mainly carbohydrates and other nutrients and components of food that cannot be digested or absorbed in the intestine [21]. These are fermented in the colon and the main carbohydrate-derived metabolites are the short-chain fatty acids (SCFA) acetate, propionate, and butyrate [22]. These metabolites play a very important role for human health, highlighting the reinforcement of the intestinal barrier, the nutrition of the protective mucosa of the intestine, improving transit in the large intestine, as well as the balance of blood glucose levels, among others [23]. However, the westernized lifestyle, characterized by a diet with high content in proteins and fats, less physical activity, greater consumption of drugs and greater stress, produces changes in our microbiota that alter its functions [24]. These alterations are known as intestinal dysbiosis and produce a general deterioration in health that manifests itself with symptoms such as diarrhea/constipation, inflammation, fatigue, allergies and even behavior changes. Intestinal dysbiosis has been shown to be behind diseases such as allergies, asthma, inflammatory bowel diseases, cardiovascular and neurodegenerative diseases, or cancer, among others [25,26,27,28,29]. Obese patients are an example of gut microbiota dysbiosis, showing a decrease in the Bacteroidetes population and a proportional increase in Firmicutes when compared to gut microbiota of non-obese individuals [30]. Also, they present a lower microbial biodiversity compared to normal weight individuals, and with an increase in plasmatic levels of lipopolysaccharides, favoring inflammatory processes. In this sense, the intestinal microbiota could be related to metabolic and trophic functions, secretion of intestinal hormones and regulators of the immune system, in addition to being involved in the regulation of fat deposits in adipose tissue [31].

As we have previously mentioned, gut microbiota plays a key role in educating and strengthening the host’s immune system to fight against multiple infections, especially those of viral origin [32]. Gut microbiota composition and its products have a positive impact on the immune system and inflammatory processes. The gut microbiota is not only able to induce anti-inflammatory responses, but also maintains the balance between the pro-inflammatory and anti-inflammatory responses to modulate the immune response against pathogens. Therefore, there is a synergism between the immune system and the gut microbiota, which helps to prevent and adequately manage different infections that may affect the host’s health. Likewise, gut microbiota-derived metabolites are also able to collaborate with the immunomodulatory and anti-viral actions of intestinal microbiota through the inhibition of viral replication and the improvement of immune response by the increase in epithelial endurance and regulatory T cell production [33,34]. Several authors have suggested the existence of direct communication between gut microbiota and lungs, known as the gut–lung axis, which allows the bidirectional transport of microbial toxins and metabolites synthesized by gut and lung microbiota through the lymphatic and circulatory system [35,36,37]. Therefore, gut microbiota disorders would be responsible for worsening respiratory outcomes such as in a SARS-CoV-2 infection [35,36,37,38,39,40]. Thus, gut microbiota composition may explain, at least in part, the susceptibility, intensity, and prognosis of the infection by SARS-CoV-2 in human patients, which would make gut microbiota intervention an important strategy to resolve or mitigate this pathology.

## 3. Effects of Gut Microbiota on the Development and Prognosis of COVID-19

Some clinical studies have shown that the gut microbiota in COVID-19 patients is significantly different from that of healthy controls [41]. The level of gut dysbiosis has been reported to be directly proportional to the severity of COVID-19 and the concentrations of pro-inflammatory and pro-oxidative markers in the plasma [38,39,40] (Figure 1). 

COVID-19 patients exhibit a decreased abundance of certain beneficial bacteria, such as *Bifidobacterium* and *Lactobacillus*, and an increase in potential pathogenic bacteria, such as *Streptococcus* and *Clostridium* [42,43]. On the other hand, *Faecalibacterium prausnitzii*, the main butyrate-producing bacteria, has been found to have a negative correlation with symptoms such as chest tightness after physical activity and cough [44]. Another study revealed a higher abundance of *Ruminococcus gnavus*, a bacterium associated with intestinal inflammation, as well as lower levels of *Faecalibacterium prausnitzii* in the gut microbiota of COVID-19 patients [42]. A recent systematic review identified the main alterations in the gut microbiota of COVID-19 patients, including a higher abundance of *Bacteroides, Clostridium, Rothia,* and *Streptococcus*, and a lower presence of *Bifidobacterium, Ruminococcus, Alistipes, Blautia, Eubacterium, Faecalibacterium*, and *Roseburia* compared to non-infected subjects [41]. Some of these alterations have been associated with COVID-19 severity and poor prognosis [41].

It is important to note that, following the resolution of the acute phase, some patients may gradually restore their gut microbiota composition to its pre-infection state. However, for others, their gut microbiota may undergo long-term changes, transitioning to a less healthy profile [45]. It has been reported that microbiota dysbiosis was not restored to normal levels even 6 months after SARS-CoV-2 infection [46]. Moreover, several studies have indicated that gut dysbiosis observed after the resolution of the infection could be involved in the persistence of some symptoms in fully recovered patients, a phenomenon known as “long COVID-19” [47,48,49]. In this sense, Zhou et al. [44] discovered that patients with persistent COVID-19 have reduced bacterial diversity three months after discharge, and the bacterium *Intestinibacter bartlettii* showed a positive correlation with persistent symptoms like fatigue, myalgia, and anorexia. Additionally, the authors observed that persistent respiratory and neuropsychiatric symptoms were correlated with opportunistic intestinal pathogens, while patients without persistent COVID-19 had a richer presence of butyrate-producing bacteria, including *Bifidobacterium pseudocatenulatum* and *Faecalibacterium prausnitzii*. Therefore, it is advisable not to leave the restoration of the gut microbiota to chance.

Although the global population has been severely affected by COVID-19, it has been identified that elderly patients or those with diabetes, hypertension, asthma, or cancer have worse clinical outcomes than the general population (Figure 2). In this sense, it is well-known that elderly patients or those with diabetes, hypertension, asthma, and cancer are associated with less gut microbiota diversity, which would be directly related to the higher morbidity and mortality observed in these risk groups compared to non-risk groups [32,50,51]. Smokers are also considered a risk population for bad prognosis of COVID-19 since cigarette smoking may alter gut microbiota abundance and composition [52]. Similarly, it has also been proposed that some geographic regions with high levels of air pollution (because of multiple polluting human activities), showed one of the highest morbidity and mortality rates of COVID-19 during the pandemic, at least in part, since air pollution constitutes an environmental factor which also may significantly contribute to gut dysbiosis [53]. That is, individuals with high susceptibility to acquire the infection by SARS-CoV-2 and to have a bad prognosis tend to have gut dysbiosis, which leads to a pro-inflammatory and pro-oxidative general state that debilitates the immune function of infected individuals [54] (Figure 2).

Oral antibiotics can have acute or long-term effects as they eliminate antiproteolytic bacteria and increase proteolytic activities in the colon, impairing the gut barrier and causing intestinal inflammation [55,56]. In some cases, the use of antibiotics to prevent complications during COVID-19 has been detected, which may lead to the exacerbation of gut dysbiosis and have a negative impact on the course of the viral infection [57,58]. 

## 4. Effects of COVID-19 on Microbiota Alterations

The way SARS-CoV-2 can affect the gut microbiome independently of hospitalization and treatment has been investigated with K18-hACE2 mice (K18-ACE2tg mice) that overexpress ACE2 [59]. These mice are characterized by the development of severe respiratory disease in a virus dose-dependent manner, resembling that which occurs in COVID-19 patients. The investigators have demonstrated that SARS-CoV-2 infection induces gut microbiota dysbiosis in mice, which correlated with abnormalities on Paneth cells and Goblet cells, and with the increment of epithelial barrier permeability. The investigators observed a decrease in gut microbiota diversity of infected mice [59]. 

Several mechanisms could explain how COVID-19 can impact the gut microbiota. Firstly, direct infection of colonocytes by the SARS-CoV-2 virus could lead to alterations in the intestinal barrier and bacterial composition through the stimulation of pathogen-associated molecular patterns (PAMPs) receptors. Invasion of SARS-CoV-2 can activate various pattern recognition receptors, including Toll-like receptors (TLRs), RIG-I-like receptors (RLRs), and NOD-like receptors (NLRs) which are recognized by innate immune cells such as dendritic cells and macrophages, triggering the release of pro-inflammatory cytokines via the nuclear factor kB (NF-kB) and JAK/STAT signaling pathways [60]. This underlying inflammatory state induced by the infection can have a negative impact on the gut microbiota. As a result, an imbalanced gut microbiota may occur, characterized by an increase in the abundance of the Proteobacteria phylum (including families such as *Enterobacteriaceae*) and a decrease in commensal bacteria such as *Eubacterium* and *Roseburia* genera (Figure 3).

Furthermore, oxidative stress has been considered as another important factor affecting COVID-19 [61]. In COVID-19 patients, 8-isoprostaglandin F2 alpha levels, considered as a marker of oxidative tissue damage, were elevated [62], and in a recent study [38], COVID-19 patients showed significantly higher values of hydrogen peroxide (H_2_O_2_) and NADPH oxidase 2 (NOX2) activity in serum, as well as TNF-α and IL-6. These results were accompanied by an increase in the serum levels of zonulin, an indirect marker of the permeability of the intestinal mucosa [63]. On the contrary, the flow-mediated dilation measured in the brachial artery and the availability of nitric oxide decreased significantly in these patients, generating endothelial dysfunction [38]. As already described, SARS-CoV-2 depends on the ACE2 receptor protein to enter the intestinal cells of the host [64], and ACE2 has, among others, the function of degrading angiotensin II (AngII). Consequently, Ang II levels increase significantly, and therefore the levels of superoxide anions and other reactive oxygen species also increase [65]. This will result in a high oxidative stress state in patients, which will also affect intestinal cells exacerbating the progression of COVID-19 [66] (Figure 3). 

In addition, the way SARS-CoV-2 can impact the gut microbiota involves the disruption of the intestinal physical barrier by the virus, leading to impaired intestinal permeability and the destruction of tight junction proteins like occludin, junctional adhesion molecule-A (JAMA), and claudin-A. This disruption can facilitate the leakage of opportunistic microorganisms into the bloodstream, resulting in systemic inflammation [67]. Another potential mechanism involves the role of the neutral amino acid transporter B0AT1 in regulating the intestinal amino acid metabolism. ACE2 also acts as a chaperone for B0AT1 in the small intestine [68]. Glutamine and tryptophan are substrates for B0AT1 and promote the formation of tight junctions via the mTOR signaling pathway. Therefore, during COVID-19 infection, the interaction between ACE2 and B0AT1 could lead to the down-regulation of B0AT1 expression on the membranes of intestinal cells. This downregulation could reduce the formation of antimicrobial peptides by Paneth cells and create a favorable environment for the growth of opportunistic bacteria in the intestine, thus establishing a dynamic vicious cycle [69] (Figure 3).

As we previously mentioned, the state of dysbiosis favored by SARS-CoV2 infection decreases the abundance of butyrate-producing bacteria. Butyrate can inhibit histone deacetylase activity, which therefore increases histone acetylation, chromatin opening and influences on gene regulation. In this sense, the NF-κB signaling pathway has less influence on the transcription of genes that encode for pro-inflammatory cytokines [45]. Thus, butyrate can differentiate naive T cells to Treg cells, which play a central role in suppressing inflammatory responses through the inhibition of histone deacetylase. In fact, this T-cell subset have the capacity to release a great variety of anti-inflammatory cytokines and prevent tissue damage [70]. Consequently, a lower butyrate concentration is reflected in lesser anti-inflammatory and immunoregulatory capacities at systemic level [71].

## 5. Underlying Mechanisms of the Gut Microbiota Effects on COVID-19 Onset and Evolution

Many mechanisms may relate the gut microbiota dysbiosis to the development and clinical outcomes of COVID-19. COVID-19 patients show a diminution of *Lactobacillus* species that may produce lactic acid because of carbohydrate fermentation, and the inactivation of viruses such as SARS-CoV-2 by pH changes. Interestingly, some strains of *L. casei*, *L. plantatum* and *L. fermentum* produce bacterial metabolites that attack the spike glycoprotein of other coronaviruses, preventing their transmission. In addition, bacteria belonging to the phylum Firmicutes can synthesize high amounts of butyrate, which is the main energy source of colonic epithelial cells responsible to produce antiviral compounds that would prevent SARS-CoV-2 invasion and multiplication [54,72]. Moreover, gut microbiota improves antiviral immunity by increasing the amount and stimulating the function of immune cells and interferon synthesis, among many other mechanisms [73]. During viral infections, gut microbiota modulates the macrophage and neutrophil activity against respiratory viruses, such as SARS-CoV-2, and can improve respiratory defenses through the induction of granulocyte-macrophage colony-stimulating factor (GM-CSF) signaling, thus promoting pathogen destruction and elimination by alveolar macrophages [74]. It has also been determined that host cellular microRNAs modulated by human gut microbiota may regulate viral replication and host gene expression implicated in the development and progression of COVID-19 [34]. 

Likewise, another study showed that low levels of *Collinsella* would predict elevated COVID-19 mortality rates [75]. One possible explanation of this phenomenon may be the fact that *Collinsella* produces ursodeoxycholate. This bile acid is able to inhibit the binding between SARS-CoV-2 and its receptor ACE2, as well as to reduce the levels of pro-inflammatory cytokines (TNF-α, IL-2, IL-1β, IL-6 and IL-4), causing anti-inflammatory, antioxidant and anti-apoptotic effects and reducing the alveolar fluid accumulation observed during the acute respiratory distress syndrome associated with COVID-19 [75]. Regarding ACE2, the receptor for SARS-CoV-2 may have a dual function in COVID-19 disease: worsening infection by increasing SARS-CoV-2 cell entry or attenuating inflammation and organ failure by balancing the ACE/ACE2 axis [76,77,78,79]. It has been reported that gut microbiota can up- or down-regulate the ACE2 expression in colonic cells and enterocytes of the small intestine. Thus, high or low COVID-19 infectivity could be influenced and dependent on gut microbiota composition [80,81,82,83].

## 6. Modulation of Gut Microbiota: Approaches in Prevention and Intervention in COVID-19

Accumulating evidence suggests that diet plays a crucial role in modulating gut microbiota composition. Nutritional disorders, characterized by excessive intake of low-quality foods, can lead to gut dysbiosis by promoting the growth of unhealthy microbiota. This, in turn, induces chronic inflammation that contributes to the immune impairment and hyperinflammation observed during COVID-19 infection. Dietary fats have been found to alter the gut microbiota by increasing the abundance of Gram-negative bacteria in the gut and intestinal permeability. This can worsen the exaggerated inflammatory response seen in COVID-19 by facilitating bacterial translocation and the release of pro-inflammatory endotoxins (lipopolysaccharides) into the systemic circulation [84,85,86,87].

In COVID-19 patients, high concentrations of the SARS-CoV-2 virus are often observed in the gut. The increased intestinal permeability mentioned earlier may contribute to the leakage of the virus into the circulation, leading to systemic distribution and potentially causing multiorgan complications [88]. Therefore, controlling the amount, frequency, and quality of ingested foods in COVID-19 patients becomes crucial to prevent nutritional disorders and reduce the severity of the disease [89,90] (Figure 4). 

A study has even suggested that the consumption of a healthy homemade diet during the pandemic confinement may have enriched the beneficial gut microbiota in many individuals, resulting in a better prognosis for COVID-19 [91]. One study conducted on 95 healthy adults showed that individuals who followed a balanced diet with a high daily consumption of vegetables, fruits, and nuts had a significantly lower risk (approximately 86% lower) of developing COVID-19 compared to those with an imbalanced diet and lower intake of these products. This difference in risk could be attributed to variations in gut microbiota diversity and composition between the two groups [92]. Thus, the prognosis of COVID-19 not only depends on gut microbiota composition but also on the available substrates that influence the gut microbiota metabolism. In this context, a study determined that high-glucose diets based on starch, galactose, and sucrose were strongly associated with a poor disease prognosis, while low-glucose diets primarily composed of whole grains were strongly correlated with a favorable COVID-19 prognosis [93]. Similarly, traditional Chinese medicines, mainly based on plant extracts, may also be useful in the treatment of COVID-19 due to their ability to positively modify the gut microbiota [94]. For example, Zhengganxifeng decoction is known to help prevent gut dysbiosis while preserving intestinal barrier integrity and increasing the abundance of SCFA-producing bacteria, which can be beneficial in the prevention and treatment of COVID-19 [95].

A direct correlation has been found between a high consumption of fermented foods (e.g., cabbage) and a low COVID-19 death rate. One possible explanation is that these foods contain significant amounts of sulforaphane precursors, which are important natural activators of nuclear factor (erythroid-derived 2)-like 2 (Nrf2), one of the most powerful antioxidants in humans. Nrf2 can help counteract the pro-oxidative mechanisms induced by the imbalance in the ACE/ACE2 axis during COVID-19. Furthermore, fermented vegetables contain abundant *Lactobacillus*, which are also known to activate Nrf2 [96].

SCFAs produced through the fermentation of dietary fiber by gut microbiota have been shown to have anti-inflammatory effects and promote tolerance and resistance to viral pathogens [97]. In fact, butyrate administration has been proposed as supportive therapy in COVID-19 treatment [98]. One study revealed that SCFAs derived from the gut microbiota metabolism can reduce respiratory and intestinal viral loads by down-regulating ACE2 and enhancing adaptive immunity through free fatty acid (FFA) receptors 3 (GPR41) and 2 (GPR43) in male animals. SCFAs directly interact with these G protein-coupled receptors (GPR41 and GPR43) [99]. Additionally, the same study proposed a novel role for the gut microbiota in reducing hypercoagulation associated with COVID-19 by limiting the proliferation of megakaryocytes and platelet turnover through the Sh2b3-Mpl axis. Sh2b3 is an adaptor protein that regulates various signal transduction pathways, including the thrombopoietin (TPO) pathway, and plays a crucial role in regulating the coagulation response [99]. 

Another study demonstrated that treatment of human endothelial progenitor cells with docosahexaenoic acid (DHA) and eicosapentaenoic acid (EPA), two N-3 polyunsaturated fatty acids derived from the diet, effectively inhibited the expression of IL-6 and impeded the entry of SARS-CoV-2 into these cells. This mechanism was mediated by the gut microbiota-derived metabolite trimethylamine-N-oxide (TMAO). The stimulation of TMAO induced by DHA and EPA leads to the inactivation of the NF-κB signaling pathway, reduced expression of ACE2 and transmembrane serine protease 2, as well as inactivation of the MAPK/p38/JNK signaling pathways and down-regulation of microRNA (miR)-221. These findings provide further evidence that gut microbiota and its metabolites may act as regulatory mediators of cytokine production and cellular infection mechanisms by SARS-CoV-2 [100].

It has been suggested that nutritional manipulation of the gut microbiota through interventions such as the consumption of probiotics, prebiotics, or symbiotics, may reduce inflammation and reinforce the immune response during COVID-19 infection, helping prevent or attenuate the severity of this viral infection [54,72,101]. This has been supported by clinical studies in which COVID-19 patients who consumed probiotics continuously since their disease diagnosis had a shorter clinical course, milder symptoms, and fewer digestive symptoms compared to those who did not consume probiotics [102,103]. Another clinical study found that the administration to COVID-19 patients of a novel symbiotic formula, composed of *Bifidobacterium* strains, xylooligosaccharide, resistant dextrin, and galactooligosaccharides, during 4–5 weeks of treatment accelerated SARS-CoV-2 antibody (IgG) formation, decreased nasopharyngeal viral load and pro-inflammatory markers (IL-6, MCP-1, M-CSF, TNF-α, and IL-1RA), and restored the gut microbiota dysbiosis [104]. In addition, some dietary microbes that can beneficially modulate the host’s gut microbiota have shown potent antiviral actions against other types of coronaviruses, making them potentially useful for the treatment of COVID-19 [105,106]. Probiotics have a strong impact on the immune system by improving the production of type I interferons, natural killer cells, T cells, antigen-presenting cells, and specific antibodies, which can act at the respiratory level [107]. The effects of probiotics on gut microbiota composition may also have therapeutic actions in COVID-19 by regulating different mechanisms involved in its pathogenesis, including SARS-CoV-2 entry through ACE2 receptors, immune response activation and immunomodulation mediated by NLR family pyrin domain-containing 3 (NLRP3), immune cell recruitment responsible for associated pulmonary and cardiovascular damage, and impairment of metabolic pathways related to COVID-19 prognosis [108]. 

On the other hand, fecal microbiota transplantation (FMT) from healthy to infected patients has been proposed as a promissory therapy in the treatment of COVID-19, since this procedure has shown the ability to induce a significant change in the gut microbiota composition towards species of bacteria that are known to induce anti-inflammatory effects (e.g., IL-10 production) in other viral infections [35,36,37]. Likewise, studies in COVID-19 patients have shown that FMT improved part of the gastrointestinal symptoms, blood immunity markers (increasing memory B and double positive T cells), as well as gut microbiota composition with a significant increase in *Bifidobacterium* and *Faecalibacterium* [109]. However, more preclinical, and clinical studies would be necessary to establish a causal relationship between FMT and the regulation of the intestinal microbiota in these patients and understand the impact of altered gut microbiota on post-infection recovery. 

Some studies have also suggested that the gut microbiota can improve the efficacy of COVID-19 vaccines and reduce associated adverse effects. This is because the success and safety of vaccination largely depend on the appropriate immune state of vaccinated individuals, enabling them to develop proper immunity against SARS-CoV-2 and be prepared for future reinfections [110,111,112,113]. In fact, it has been reported that the T cell response induced by COVID-19 mRNA vaccines is inversely correlated with the expression of activation protein 1 (AP-1). Moreover, AP-1 is positively correlated with the gut microbiota’s fucose/rhamnose degradation pathway [114]. Furthermore, it is known that the immune response to COVID-19 vaccines tends to decrease with age due to gut dysbiosis, usually associated with aging [115].

Interestingly, not only gut microbiota, through bacteria, may modulate host immunity and affect the severity of SARS-CoV-2 infection, but also gut virome (viruses) and fungi (mycobiome) could do it [116,117,118,119]. The human gut virome consists of numerous resident viruses that may be crucial in immunoregulation and other related aspects, including the pathophysiology of various diseases, including COVID-19 [119]. In this context, Lu et al. discovered that the DNA of the crAss-like phages family was increased in COVID-19 patients compared to healthy controls. Furthermore, the viral family *Tectiviridae* and the bacterial family *Bacteroidaceae* exhibited significant co-increases during the course of SARS-CoV-2 infection, suggesting a linked evolution between the virome and bacteriome [117]. Similarly, it has also been observed that there are differences in the virome due to SARS-CoV-2 infection, and this dissimilarity may contribute to the severity of the disease and the recovery process [116]. Scientific evidence regarding mycobiome is more limited, but some studies are emerging. In this regard, Reinold et al. have observed a reduced diversity, richness, and uniformity and an increase in the relative abundance of the Ascomycota phylum in severe/critical COVID-19 patients compared to non-severe cases. This study shows that the dominant fungal species were variable between patients, even within patient groups. However, in contrast to what has been described in the microbiota, where no major phylum has been found, Ascomycota phylum was present with a relative abundance > 75% in most severe/critical COVID-19 patients [118]. It has also been reported that hospitalized COVID-19 patients exhibit an increased presence of fungal pathogens from the *Candida* and *Aspergillus* genera (both Ascomycota phylum) compared to non-infected individuals. Since some fungal genera found increased in SARS-CoV-2 patients can produce antimicrobial metabolites that affect bacterial proliferation, the mycobiome could further induce gut microbiota dysbiosis [118]. Unstable gut mycobiomes persisted in a subset of COVID-19 patients for up to 12 days after the clearance of SARS-CoV-2 from the nasopharynx [45]. Therefore, therapeutic strategies aimed at targeting the gut virome and mycobiome could be beneficial in combating this viral disease. Further studies are necessary to determine whether the proliferation of both viral and fungal microorganisms is a cause or a consequence of SARS-CoV-2 infection itself and whether they are related to gut microbiota dysbiosis.

## 7. Conclusions and Prospects

The COVID-19 pandemic has taught us numerous lessons, and perhaps one of the most significant is the recognition of the crucial role the gut microbiota plays in its pathophysiology and prognosis, which had been previously underestimated. It has also highlighted the importance of maintaining good intestinal health to prevent the devastating consequences of a disease that will be remembered by all of mankind. Advancing our understanding of the close and bidirectional relationship between COVID-19 and the gut microbiota, as well as its implications for nutritional and therapeutic management (with both natural and synthetic drugs), holds great importance. Another essential aspect to be addressed should be the further study of the intestinal virome and mycobiome and the recognition of their key participation in the COVID-19 pandemic (beyond the assessment of harmful/beneficial bacteria populations which are better studied), two novel and very interesting concepts that have recently emerged within the complex system of the gut microbiota. This knowledge will enable us to be better prepared for future pandemics, thereby avoiding the repetition of past mistakes.

## Figures and Tables

**Figure 1 ijms-24-12249-f001:**
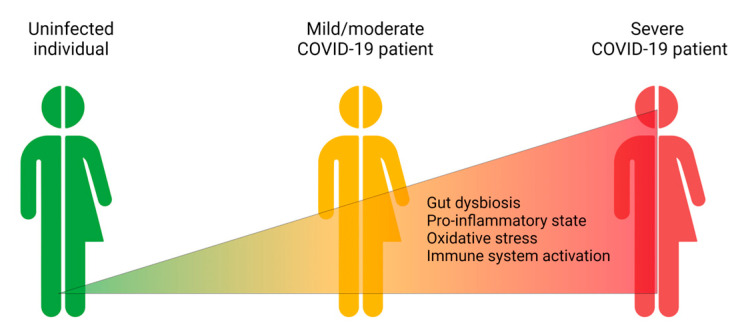
SARS-CoV-2 severity is associated with increased gut dysbiosis, a pro-inflammatory and pro-oxidant state, and an activated immune response. Created with BioRender.com.

**Figure 2 ijms-24-12249-f002:**
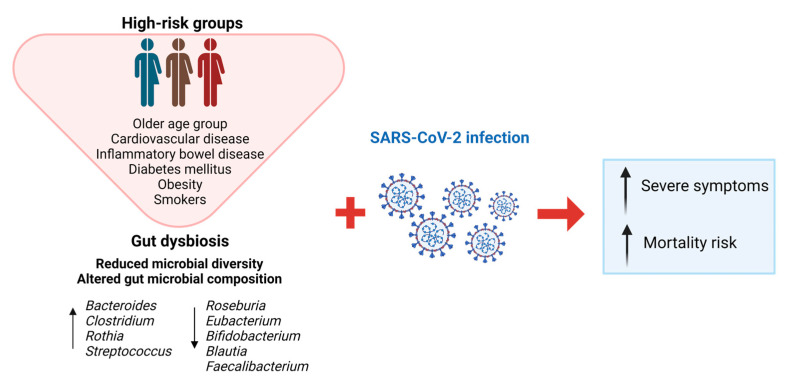
Characteristics of gut dysbiosis in high-risk COVID-19 patients and main symptoms associated with SARS-CoV-2 infection. Created with BioRender.com.

**Figure 3 ijms-24-12249-f003:**
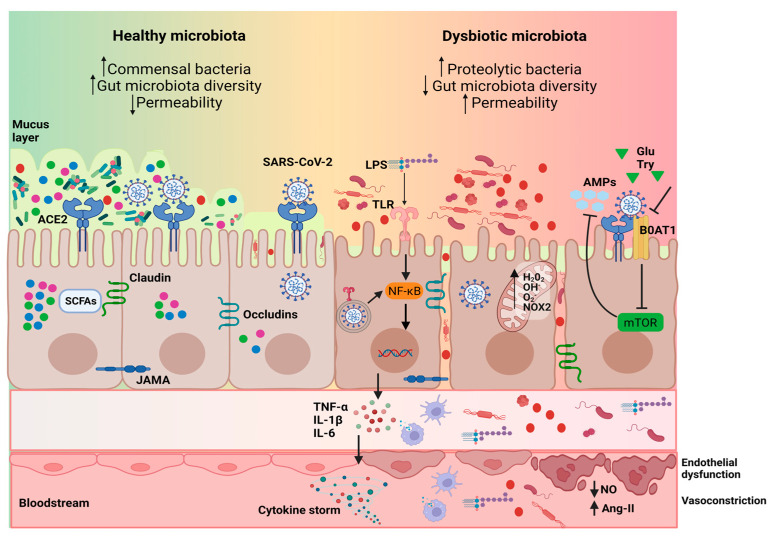
Possible mechanisms by which SARS-CoV-2 infection induces dysbiosis of the gut microbiota are as follows: (1) activation of Toll-like receptors (TLRs) triggers the release of pro-inflammatory cytokines through the nuclear factor κB (NF-κB) signaling pathways; (2) the virus destroys tight junction proteins, leading to the disruption of the intestinal physical barrier and increased permeability; (3) down-regulation of angiotensin-converting enzyme 2 (ACE2) and B0AT1 expression in intestinal epithelial cells decreases the production of antimicrobial peptides (AMPs), facilitating the growth of pathogens; (4) mitochondrial dysfunction and high oxidative stress induced by the virus infection occurs. This infection-induced underlying inflammatory and oxidative state induces microbiota dysbiosis characterized by a decrease in the abundance of commensal bacteria such as the genera *Eubacterium* and *Faecalibacterium*, and an increase in the abundance of the genus *Enterococcus* and proteolytic bacteria. These bacteria or the lipopolysaccharides (LPS) that they contain, might translocate into the circulatory system and be recognized by innate cells, resulting in increased pro-inflammatory responses and endothelial dysfunction. This creates a vicious circle that exacerbates the progression of COVID-19. Ang-II: angiotensin II; BoAT1: sodium-dependent neutral amino acid transporter; Glu: Glutamine; IL-6: interleukin 6; LPS: lipopolysaccharides; mTOR: mechanistic target of rapamycin; NO: nitric oxide; SCFAs: short-chain fatty acids; TNF-α: tumoral necrosis factor alpha; Try: tryptophan. Created with BioRender.com.

**Figure 4 ijms-24-12249-f004:**
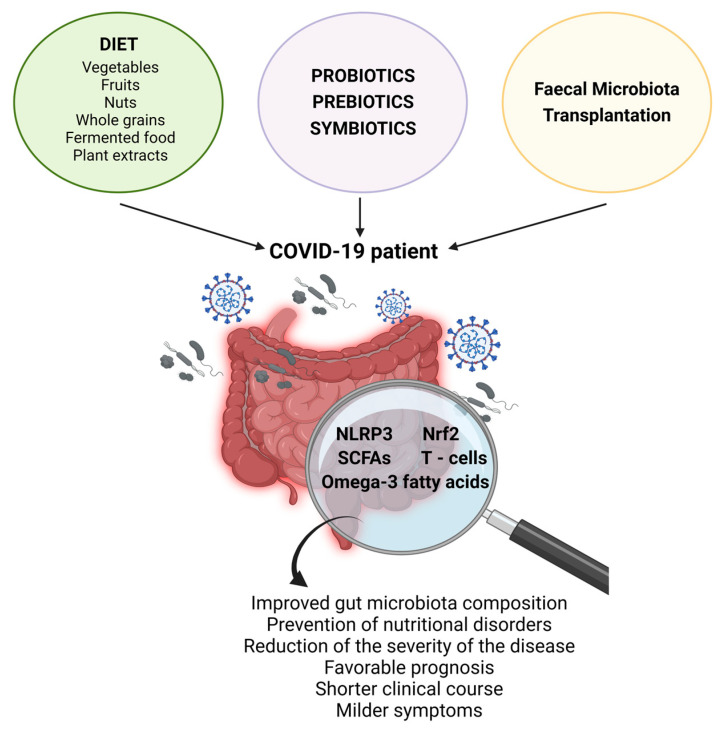
Possible nutritional and therapeutic interventions on gut microbiota and their beneficial effects in COVID-19 patients. Created with BioRender.com.

## Data Availability

Not applicable.

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
