# Peer review of "Gut Microbiota Dysbiosis in COVID-19: Modulation and Approaches for Prevention and Therapy"

_ijms, 2023, doi:10.3390/ijms241512249_

Round 1

Reviewer 1 Report

Very interesting paper, I would ask you just a few clarifications to make.

How long does the alterations due to Covid become established? 

Previously the stays in Covid positive patients were very long, now increasingly shorter, so I would ask you to expose well in what kind of patients and after how long the alterations you describe appear.

The second of the two figures you have included is really well done, very simple and explanatory, in this regard I would suggest including a couple more, or even simple diagrams.

For example, one that summarizes in what kind of patients and how soon this dysbiosis appears and what effects it produces and one that summarizes the therapeutic interventions proposed in Chapter 6.

Author Response

We want to thank the reviewer for his/her comments. We think that the manuscript is now more complete and comprehensible for the potential reader based on the comments of the two reviewers. In the responses to the reviewers, we have tried to harmonize the comments of all of them.

  1. How long does it take to establish the alterations due to Covid? Before, the stays in Covid-positive patients were very long, now they are increasingly shorter, so I would ask you to explain in what type of patients and after how long do the alterations you describe appear.

Author’s reply:

Many studies have investigated the impact of the SARS-CoV-2 infection on gut microbiota composition of patients from an asymptomatic, and mild to severe clinical presentation of COVID-19, and after recovery [reference 41: Farsi Y et al. Front Cell Infect Microbiol. 2022;12:804644; reference 45: Zuo T et al. Gastroenterology. 2020;159(3):944-955.e8). There are many differences among the studies attributed to many factors including the age, comorbidities, gastroenterology symptoms, clinical evolution or treatments during hospitalization of the patients (when they have been documented). In addition, most of them have been done with case-control or cohorts with a transversal design. Despite some differences among studies, gut microbiota composition of recovered COVID-19 patients persisted different to that found in non-infected control subjects even after 6 months after infection [reference 46: Simadibrata DM et al. J Dig Dis. 2023;24(4):244-261]. Few studies that have investigated the gut microbiota composition on patients with persistent symptoms in post-acute COVID-19 syndrome (known as long COVID) have been reported that microbiota dysbiosis was not restored to normal levels even after 12 months after SARS-CoV-2 infection [reference 46: Simadibrata DM et al. J Dig Dis. 2023;24(4):244-261]. Thus, we need longitudinal studies that explore changes in gut microbiota composition in COVID-19 patients from healthy status after the recovery, in particular, for the identification of a gut microbiota profile that could protect against COVID-19 infection. This information has been included in this new version along the text.

  1. The second of the two figures that you have included is very well done, very simple and explanatory, in this sense I suggest you include a couple more, or even simple diagrams. For example, one that summarizes in what type of patients and when this dysbiosis appears and what effects it produces and another that summarizes the therapeutic interventions proposed in Chapter 6.

Author’s reply:

At the request of the reviewer, we added two figures. The first (Figure 2) summarizes the high-risk patient groups and the characteristics of gut dysbiosis in these individuals, which could contribute to increased susceptibility to COVID-19. In addition, we show the changes in the diversity of the gut microbiota due to SARS-CoV-2 infection, as well as more significant symptoms of the syndrome.

The second figure (Figure 4) summarizes the possible interventions in COVID-19 based on the modulation of the composition of the intestinal microbiota. Numerous studies suggest the role of diet, probiotics, prebiotics and symbiotics, and fecal microbiota transplantation in improving the antiviral capacity and the severity and symptoms of COVID-19, through the modulation of the gut microbiota.

Author Response

We want to thank the reviewer for his/her comments. We think that the manuscript is now more complete and comprehensible for the potential reader based on the comments of the two reviewers. In the responses to the reviewers, we have tried to harmonize the comments of all of them.

  1. Kindly provide a detailed explanation of the abbreviations used in the initial sentence.

Author’s reply:

At the reviewer’s request, we added a detailed explanation of the abbreviations used in the initial sentence, as follows: “…COVID-19 (by the acronym of Coronavirus Disease of 2019) is a respiratory disease caused by a novel coronavirus (SARS-CoV-2, by the abbreviation of Severe Acute Respiratory Syndrome Coronavirus 2) that continues to affect millions of people worldwide…”.

  1. Section 2. sentence “This data is important considering that most of the studies are carried out on fecal samples, which mainly represent the distal intestinal (colonic) microbiota.” Please augment the statement with additional information and include relevant references to support it.

Author’s reply:

We augmented that statement with additional information and included relevant references to support it, as follows: “… This data is important considering that most of the studies are carried out on fecal samples, which mainly represent the distal intestinal (colonic) microbiota. Fecal specimens are naturally collected, may be sampled repeatedly, and is a less invasive procedure than others, such as intestinal biopsy sample collection, luminal brushing, and intestinal fluid aspirate, which are not suitable methods for healthy people [reference 7: Ticinesi A et al. The impact of intestinal microbiota on bio-medical research: definitions, techniques and physiology of a "new frontier". Acta Biomed. 2018 Dec 17;89(9-S):52-59. doi: 10.23750/abm.v89i9-S.7906.; reference 8: Tang Q et al. Current Sampling Methods for Gut Microbiota: A Call for More Precise Devices. Front Cell Infect Microbiol. 2020 Apr 9;10:151. doi: 10.3389/fcimb.2020.00151]. However, it is important to clarify that stool samples do not represent the totality of the microbiota adhered to the intestinal epithelium and that the bacteria of the upper intestinal tracts are not correctly detected [reference 9: Zmora N. et al. Cell. 2018 Sep 6;174(6):1388-1405.e21. doi: 10.1016/j.cell.2018.08.041].

  1. Sentence:” The conformation of the microbiota in each subject is unique and is determined by a multitude of factors such as the way we are born, whether it is by vaginal delivery or caesarean section, the type of diet, drug intake (mainly antibiotics), and even if one lives in an urban or rural environment [7].” – I strongly suggest incorporating more references to corroborate your general statements and further strengthen the validity of your arguments.

Author’s reply:

In complete agreement with the reviewer, we incorporated the following references [10-13] to corroborate and strengthen those general statements: “…The conformation of the microbiota in each subject is unique and is determined by a multitude of factors such as the way we are born, whether it is by vaginal delivery or caesarean section [10: Zhang C et al. The Effects of Delivery Mode on the Gut Microbiota and Health: State of Art. Front Microbiol. 2021 Dec 23;12:724449. doi: 10.3389/fmicb.2021.724449.], the type of diet [11: Leeming ER et al. Effect of Diet on the Gut Microbiota: Rethinking Intervention Duration. Nutrients. 2019 Nov 22;11(12):2862. doi: 10.3390/nu11122862.], drug intake (mainly antibiotics) [12: Ramirez J et al. Antibiotics as Major Disruptors of Gut Microbiota. Front Cell Infect Microbiol. 2020 Nov 24;10:572912. doi: 10.3389/fcimb.2020.572912.], and even if one lives in an urban or rural environment [13: Tyakht AV et al. Rural and urban microbiota: To be or not to be? Gut Microbes. 2014 May-Jun;5(3):351-6. doi: 10.4161/gmic.28685.,…”.

  1. “The more different types of bacteria in the gut and the more evenly distributed they are, the greater the diversity. A diverse microbiome can perform many more functions, making the whole system more stable. In fact, a diverse and balanced intestinal microbiota ensures the correct functioning of the digestive tract, strengthens the immune system, and improves metabolism.” The entire section lacks any supporting references. To enhance its credibility, please include relevant references and additional text to substantiate the information presented.

Author’s reply:

We incorporated the following references to support and enhance the credibility of this section: “…The more different types of bacteria in the gut and the more evenly distributed they are, the greater the diversity. A diverse microbiome can perform many more functions, making the whole system more stable. In fact, a diverse and balanced intestinal microbiota ensures the correct functioning of the digestive tract, strengthens the immune system, and improves metabolism [reference 18: Ho CT et al. Editorial note: Gut microbiota and health. J Tradit Complement Med. 2023 Mar 13;13(2):105-106. doi: 10.1016/j.jtcme.2023.03.004.; reference 19: Lozupone CA et al. Diversity, stability and resilience of the human gut microbiota. Nature. 2012 Sep 13;489(7415):220-30. doi: 10.1038/nature11550.]…”.

  1. The section above reference 12 requires more supporting references. Although the statements appear to be well-known facts, it is crucial to connect them with published references to reinforce their credibility and align with established research in the article.

Author’s reply:

At the reviewer’s request, we incorporated the following references to reinforce the credibility of this section: “…Gut microbiota performs a wide variety of biochemical and physiological functions that influence host metabolism [20: Rowland I et al.  Gut microbiota functions: metabolism of nutrients and other food components. Eur J Nutr. 2018 Feb;57(1):1-24. doi: 10.1007/s00394-017-1445-8.]. The microorganisms that compose it have various enzymes that make it possible to transform mainly carbohydrates and other nutrients and components of food that cannot be digested or absorbed in the intestine [21: Vernocchi P et al. Gut Microbiota Metabolism and Interaction with Food Components. Int J Mol Sci. 2020 May 23;21(10):3688. doi: 10.3390/ijms21103688.]. These are fermented in the colon and the main carbohydrate-derived metabolites are the short-chain fatty acids (SCFA) acetate, propionate, and butyrate [22]…”.

  1. Dear authors, I believe that the statements in section 2 need to be expanded and supported with more recent references and additional text. Some of the literature sources cited are quite old, and there might be more up-to-date data available to validate your points. Moreover, several statements lack proper literary sources, and it would be beneficial to reference them to strengthen the article's credibility.

Author’s reply:

At the reviewer’s request, we incorporated more recent references and additional text to reinforce the credibility of section 2.

  1. Fusco W et al. Short-Chain Fatty-Acid-Producing Bacteria: Key Components of the Human Gut Microbiota. Nutrients 2023, 15, doi:10.3390/nu15092211.
  2. Hou K. et al. Microbiota in health and diseases. Signal Transduct Target Ther 2022, 7, 135, doi:10.1038/s41392-022-00974-4.
  3. Clemente JC et al. The impact of the gut microbiota on human health: an integrative view. Cell 2012, 148, 1258-1270, doi:10.1016/j.cell.2012.01.035.
  4. Hemmati M et al. Importance of gut microbiota metabolites in the development of cardiovascular diseases (CVD). Life Sci 2023, 329, 121947, doi:10.1016/j.lfs.2023.121947.
  5. Verhaar BJH et al. Gut Microbiota Composition Is Related to AD Pathology. Front Immunol 2021, 12, 794519, doi:10.3389/fimmu.2021.794519.
  6. Wang M et al. The role of the gut microbiota in gastric cancer: the immunoregulation and immunotherapy. Front Immunol 2023, 14, 1183331, doi:10.3389/fimmu.2023.1183331.
  7. Yao C et al. Significant Differences in Gut Microbiota Between Irritable Bowel Syndrome with Diarrhea and Healthy Controls in Southwest China. Dig Dis Sci 2023, 68, 106-127, doi:10.1007/s10620-022-07500-0.

  1. Section 3 “Some clinical studies have shown that the gut microbiota in COVID-19 patients is significantly different from that of healthy controls.” – please add some examples of this studies and references.

Author’s reply:

In response to the reviewer’s request, we incorporated the following reference to illustrate that general statement: “…Some clinical studies have shown that the gut microbiota in COVID-19 patients is significantly different from that of healthy controls [41: Farsi Y et al. Diagnostic, Prognostic, and Therapeutic Roles of Gut Microbiota in COVID-19: A Comprehensive Systematic Review. Front Cell Infect Microbiol 2022, 12, 804644, doi:10.3389/fcimb.2022.804644.

  1. “Another study revealed a higher abundance of Ruminococcus gnavus, a bacterium associated with intestinal inflammation, as well as lower levels of Faecalibacterium prausniii in the gut microbiota of COVID-19 patients.” – Please specify the study in which the term is defined. Without a reference, this sentence lacks significance and meaning.

Author’s reply:

At the reviewer’s request, we specified the study in which this term is defined: “…Another study revealed a higher abundance of Ruminococcus gnavus, a bacterium associated with intestinal inflammation, as well as lower levels of Faecalibacterium prausniii in the gut microbiota of COVID-19 patients [42: Liu Q et al. Gut microbiota dynamics in a prospective cohort of patients with post-acute COVID-19 syndrome. Gut. 2022 Mar;71(3):544-552. doi: 10.1136/gutjnl-2021-325989.]”.

  1. Section 4 - references are missing at the end of the first paragraph.

Author’s reply:

Regarding the reviewer’s observation, we added the missing reference at the end of the first paragraph in section 4, as follows: “…These mice are characterized by the development of severe respiratory disease in a virus dose-dependent manner, resembling that occurs in COVID-19 patients. The investigators have demonstrated that SARS-CoV-2 infection induces gut microbiota dysbiosis in mice, which correlated with abnormalities on Paneth cells and Goblet cells, and with the increment of epithelial barrier permeability. The investigators observed a decrease in gut microbiota diversity of infected mice [59: Bernard-Raichon L et al.  Gut microbiome dysbiosis in antibiotic-treated COVID-19 patients is associated with microbial translocation and bacteremia. Nat Commun 2022, 13, 5926, doi:10.1038/s41467-022-33395-6.]…”.

  1. Last sentence in section 4 needs reference.

Author’s reply:

In this regard, we added the missing reference at the end of the first paragraph in section 4, as follows: “…Consequently, a lower butyrate concentration is reflected in a lesser anti-inflammatory and immunoregulatory capacities at systemic level [71: Siddiqui MT, Cresci GAM. The Immunomodulatory Functions of Butyrate. J Inflamm Res. 2021 Nov 18;14:6025-6041. doi: 10.2147/JIR.S300989.]…”.

  1. Section 5. “Likewise, another study showed that low levels of Collinsella would predict elevated COVID-19 mortality rates” – Please refer to the specific study you are mentioning in your statement. Without citing the relevant study, the sentence lacks clarity and importance.

Author’s reply:

In response to the reviewer’s observation, we specified the study in which this term is defined: “…Likewise, another study showed that low levels of Collinsella would predict elevated COVID-19 mortality rates [75: Hirayama M et al. Intestinal Collinsella may mitigate infection and exacerbation of COVID-19 by producing ursodeoxycholate. PLoS One 2021, 16, e0260451, doi:10.1371/journal.pone.0260451.]…”.

  1. “Some studies have even suggested that the consumption of a healthy homemade diet during the pandemic confinement may have enriched the beneficial gut microbiota in many individuals, resulting in a better prognosis for COVID-19 [73].” – Apologies for any confusion. When using the term "some studies," please include references for those studies to support your statement effectively. Citing these references will add credibility and relevance to the mentioned studies in your context.

Author’s reply:

We changed “some studies” by “a study” in order to avoid any confusion about this sentence, as the reference what we referred was the number 91 [Rishi, P et al. Diet, Gut Microbiota and COVID-19. Indian J Microbiol 2020, 60, 420-429, doi:10.1007/s12088-020-00908-0.]

  1. “Interestingly, not only gut microbiota, through bacteria, may modulate host immunity and affect the severity of SARS-CoV-2 infection, but also gut virome (viruses) and fungi (mycobiome) could do it.” – please add references.

Author’s reply:

At the reviewer’s petition, we added the requested references, as follows: “…Interestingly, not only gut microbiota, through bacteria, may modulate host immunity and affect the severity of SARS-CoV-2 infection, but also gut virome (viruses) and fungi (mycobiome) could do it [116-119]

  1. Cao J et al. Integrated gut virome and bacteriome dynamics in COVID-19 patients. Gut Microbes 2021, 13, 1-21, doi:10.1080/19490976.2021.1887722.
  2. Lu ZH et al. Alterations in the Composition of Intestinal DNA Virome in Patients With COVID-19. Front Cell Infect Microbiol 2021, 11, 790422, doi:10.3389/fcimb.2021.790422.
  3. Reinold J et al. The Fungal Gut Microbiome Exhibits Reduced Diversity and Increased Relative Abundance of Ascomycota in Severe COVID-19 Illness and Distinct Interconnected Communities in SARS-CoV-2 Positive Patients. Front Cell Infect Microbiol 2022, 12, 848650, doi:10.3389/fcimb.2022.848650.
  4. Zuo T et al. Temporal landscape of human gut RNA and DNA virome in SARS-CoV-2 infection and severity. Microbiome 2021, 9, 91, doi:10.1186/s40168-021-01008-x.…”.

  1. “Furthermore, these authors demonstrated that a dominance of single fungal taxa is a common characteristic among severe COVID-19 patients [101].” - Dear authors, I find this aspect highly intriguing and would greatly appreciate a more detailed explanation and description. The additional elaboration will enhance the reader's understanding and engagement with the subject matter. Thank you.

Author’s reply:

At the reviewer’s request, we have included a brief explanation on this aspect. Overall, this analysis [118: Reinold J et al. The Fungal Gut Microbiome Exhibits Reduced Diversity and Increased Relative Abundance of Ascomycota in Severe COVID-19 Illness and Distinct Interconnected Communities in SARS-CoV-2 Positive Patients. Front Cell Infect Microbiol 2022, 12, 848650, doi:10.3389/fcimb.2022.848650] shows a strong modulation of positive associations between fungal genera in SARS-CoV-2 positive in comparison to SARS-CoV-2 negative patients. However, the authors emphasize that additional studies would be necessary to provide greater clarity.

  1. I kindly recommend considering a revision of the conclusion section. While the role of the microbiota is undoubtedly significant, I believe there are other essential lessons to address. Therefore, I suggest reviewing and revising the entire section to ensure its clarity and impact. Thank you for your attention to this matter.

Author’s reply:

At the reviewer’s request, we reviewed and modified the conclusion section, as follows: “…The COVID-19 pandemic has taught us numerous lessons, and perhaps one of the most significant is the recognition of the crucial role played by the gut microbiota in its pathophysiology and prognosis, which had been previously underestimated. It has also highlighted the importance of maintaining good intestinal health to prevent the devastating consequences of a disease that will be remembered by all of mankind. Advancing our understanding of the close and bidirectional relationship between COVID-19 and the gut microbiota, as well as its implications for nutritional and therapeutic (with both natural and synthetic drugs) management, holds great importance. Another essential aspect to be addressed should be the further study of the intestinal virome and mycobiome and the recognition of their key participation in COVID-19 pandemic (beyond the assessment of harmful/beneficial bacteria populations which are better studied), two novel and very interesting concepts that have recently emerged within the complex system of the gut microbiota. This knowledge will enable us to be better prepared for future pandemics, thereby avoiding the repetition of past mistakes…”.

Round 2

Reviewer 1 Report

The changes made are fine, certainly the paper is much improved.

At this point I would add a paragraph exposing the limitations of this study, which at the end of the day remains a review and gathers various insights indicating some ideas or suggestions, but still remains article more of reflection on our knowledge gained so far with thus many limitations on it.